# miR-128 Restriction of LINE-1 (L1) Retrotransposition Is Dependent on Targeting hnRNPA1 mRNA

**DOI:** 10.3390/ijms20081955

**Published:** 2019-04-21

**Authors:** Lianna Fung, Herlinda Guzman, Evgueni Sevrioukov, Adam Idica, Eddie Park, Aurore Bochnakian, Iben Daugaard, Douglas Jury, Ali Mortazavi, Dimitrios G. Zisoulis, Irene M. Pedersen

**Affiliations:** 1Department of Molecular Biology and Biochemistry, Francisco J. AyalaSchool of Biological Sciences, University of California, Irvine, CA 92697, USA; liannaf@uci.edu (L.F.); hguzman@uci.edu (H.G.); esevriou@uci.edu (E.S.); akidica@gmail.com (A.I.); aurore.bochnakian@gmail.com (A.B.); ibenld@gmail.com (I.D.); djury@uci.edu (D.J.); dzisoulis@gmail.com (D.G.Z.); 2Department of Developmental and Cell Biology, University of California, Irvine, CA 92697, USA; eddiep@ucla.edu (E.P.); ali.mortazavi@uci.edu (A.M.)

**Keywords:** miR-128, LINE-1, retrotransposition, hnRNPA1, miRs, restriction factor

## Abstract

The majority of the human genome is made of transposable elements, giving rise to interspaced repeats, including Long INterspersed Element-1s (LINE-1s or L1s). L1s are active human transposable elements involved in genomic diversity and evolution; however, they can also contribute to genomic instability and diseases. L1s require host factors to complete their life cycles, whereas the host has evolved numerous mechanisms to restrict L1-induced mutagenesis. Restriction mechanisms in somatic cells include methylation of the L1 promoter, anti-viral factors and RNA-mediated processes such as small RNAs. microRNAs (miRNAs or miRs) are small non-coding RNAs that post-transcriptionally repress multiple target genes often found in the same cellular pathways. We have recently established that miR-128 functions as a novel restriction factor inhibiting L1 mobilization in somatic cells. We have further demonstrated that miR-128 functions through a dual mechanism; by directly targeting L1 RNA for degradation and indirectly by inhibiting a cellular co-factor which L1 is dependent on to transpose to new genomic locations (TNPO1). Here, we add another piece to the puzzle of the enigmatic L1 lifecycle. We show that miR-128 also inhibits another key cellular factor, hnRNPA1 (heterogeneous nuclear ribonucleoprotein A1), by significantly reducing mRNA and protein levels through direct interaction with the coding sequence (CDS) of hnRNPA1 mRNA. In addition, we demonstrate that repression of hnRNPA1 using hnRNPA1-shRNA significantly decreases de novo L1 retro-transposition and that induced hnRNPA1 expression enhances L1 mobilization. Furthermore, we establish that hnRNPA1 is a functional target of miR-128. Finally, we determine that induced hnRNPA1 expression in miR-128-overexpressing cells can partly rescue the miR-128-induced repression of L1′s ability to transpose to different genomic locations. Thus, we have identified an additional mechanism by which miR-128 represses L1 retro-transposition and mediates genomic stability.

## 1. Introduction

Repetitive sequences make up greater than half of the human genome, of which Long INterspaced elements-1 (LINE-1 or L1) account for approximately 17% [1,2,3]. Although the majority of L1 elements are truncated and inactive, the average human genome retains 80–100 retrotransposition-competent L1 copies. The intact L1 element is approximately 6 kilobase pairs (kb) in length. L1 harbors a 5′UTR, with promoter activity in both the sense and anti-sense directions [4,5,6]. The open-reading frames encode ORF0, ORF1 and ORF2, which are followed by a short 3′UTR. ORF1 encodes an RNA-binding protein (40 kDa) with nucleic acid chaperone activity, and ORF2 encodes an endonuclease and reverse transcriptase protein, which is 150 kDa in size [7,8,9]. ORF0 is transcribed in the antisense direction and encodes a protein. However, the mechanism of ORF0 remains unknown [10]. The L1 life cycle is dependent on replicating the L1 element using a “copy and paste” mechanism with an RNA intermediate [11,12]. Integration of L1 at new locations in the genome generates mutations that can create new genes or affect gene expression [13,14]. L1 retro-transposition has been associated with a variety of diseases including hemophilia, cancer and developmental abnormalities [2,15,16,17,18]. As a result, multiple mechanisms have evolved to regulate L1 activity. In germ cells, specific small RNA subtypes (piRNAs) efficiently counteract L1 activity [19,20]. In somatic cells, L1 transcription is effectively inhibited by DNA methylation of the L1 promoter [21,22]. In hypomethylated cell populations such as cancer cells or pluripotent stem cells, the L1 promoter is often de-repressed allowing for L1 retrotransposition [22,23,24]. Under these conditions, other mechanisms of L1 restriction are crucial, including suppression by DNA and RNA editing proteins, including AID, APOBECs and ADAR [25,26], as well as the microprocessor [27].

The human transcriptome is subjected to miR regulation, emphasizing the importance of the post-transcriptional control of gene regulation by non-coding RNA (ncRNA) in regulating multiple genetic pathways [28,29]. miRs are endogenously encoded 21–23-nucleotide (nt) RNAs that regulate the expression of mRNAs containing complementary sequences. After transcription and processing in the nucleus, the mature miR is loaded onto specific Argonaute (Ago) proteins—referred to as an miR-induced silencing complex (miRISC)—in the cytoplasm. The miRISC then binds to partially complementary mRNA sequences and mediates mRNA degradation or translational inhibition [29,30]. Complementarity between the miR (5′ position 2–7) and an mRNA target “seed” site usually results in reduced protein expression through a variety of mechanisms that involve mRNA degradation and translational repression [30,31]. 

We have previously established that miR-128 represses L1 retrotransposons in somatic cells through a dual mechanism, namely by direct targeting of L1 ORF2 mRNA and indirectly through the regulation of a required cellular co-factor, Transportin 1 (TNPO1) [32,33]. It is well established that miRs repress multiple cellular mRNAs by binding to homologous target seed sequences; the proteins of these target mRNAs frequently function in the same pathway, suggesting that miRs act to fine-tune specific cellular networks [34,35,36,37]. 

In this study, we wished to identify additional cellular targets of miR-128 that may be involved in the L1 retrotransposition pathway. Here, we report that miR-128 also represses retrotransposition by targeting a key cellular co-factor required for L1 retrotransposition, namely hnRNPA1. hnRNPA1 is one of the most abundant proteins in the nucleus and is expressed in all cell types and tissues [38]. The hnRNPA1 proteins are involved in both DNA and RNA metabolism including genomic stability and telomere binding [39,40]. hnRNPA1 is bound to poly (A) sequences of RNA in both the cytoplasm and nucleus [41] and accompanies mature transcripts through the Nuclear Pore Complex, supporting its proposed role in mRNA shuttling [42]. The nuclear localization of hnRNPA1 depends on a 38 aa long nuclear localization sequence known as M9 [43,44,45], which binds to TNPO1 to mediate shuttling through the nuclear pore complex [40,46,47,48,49,50]. It has been described that hnRNPA1 interacts with L1 ORF1p through an RNA bridge, as part of the L1-RNP complex [51]. In this study, we show that miR-128 significantly decreases hnRNPA1 protein levels, by directly binding to hnRNPA1 mRNA, and that miR-128-induced L1 restriction is partly dependent on targeting hnRNPA1. Thus, we have discovered another key player in the L1 life cycle, which is subjected to miR-128 regulation.

## 2. Results

### 2.1. Identification of miR-128 Targets Which Function As Co-Factors for L1 Retrotransposition

We have previously demonstrated that miR-128 directly targets L1 RNA and represses de novo retrotransposition and genomic integration in somatic cells [32]. Furthermore, we recently determined that miR-128 also regulates cellular co-factors, some of which L1 may be dependent on. It has previously been debated whether L1-RNP is only dependent on cell division to access host DNA [52,53,54]. However, the hypothesis could not account for how L1 integrates back into the genome in non-dividing cells such as neurons [55]. For this reason, we were excited to determine that miR-128 also targets the nuclear import factor Transportin 1 (TNPO1), resulting in reduced nuclear import of L1 Ribonuclear Protein (L1-RNP) complexes [33]. However, as TNPO1 does not contain a nuclear localization signal (NLS), many questions were left unanswered. 

In order to identify additional miR-128 targets potentially involved in the regulation of L1 retrotransposition, and add to our understanding of the L1 life cycle, we performed a screen for potential miR-128 targets utilizing DGCR8^−/−^ mouse embryonic stem cells (mESCs) (a kind gift from Dr. Blelloch). The DGCR8 knockout ES cells were generated by removing exon 3, resulting in the formation of several premature stop codons downstream of the targeted region). DGCR8 is a critical component of the microprocessor involved in processing pri-miRs into their mature forms [56]. In DGCR8^−/−^ mESC cells, pri-miR transcripts cannot be further processed by the microprocessor and are not loaded into a miRISC complex [57]. As a result, the DGCR8^−/−^ system is free of mature, biologically active canonical miRs. We transfected DGCR8^−/−^ mESCs with miR controls or miR-128 in triplicate cultures and harvested cells after 12 h in order to enrich for primary target mRNAs, as opposed to studying secondary effects of miR-128. Two replicates of each triplicate were selected and cDNA libraries were generated using the Smart-seq2 protocol [58]. The libraries were sequenced as 43 bp paired-end reads. The ultrafast universal RNA-seq aligner STAR (Spliced Transcripts Alignment to a Reference) was used to align the reads on to the mm9 genome [59]. RSEM (RNA-Seq by Expectation-Maximization) [60] was used to quantitate the gene expression and EBSeq [61] was used to identify differentially expressed genes. We then performed overlay analysis of the identified miR-128 targets with previously reported results from one proteomic screen identifying L1 ORF1p-encoded protein (ORF1p) interaction partners [51], see Appendix A. Interestingly, several members of the hnRNP (heterogeneous nuclear ribonucleoprotein) family were identified (hnRNPA1, hnRNPA2B1, hnRNPK, hnRNPL and hnRNPU) as potential miR-128 targets and ORF1p interaction partners, see Figure 1A, shown in grey. To validate the findings of the primary screen, we generated lentiviruses containing plasmids encoding miR-128, anti-miR-128 or scramble miR control. HeLa cells were transduced and selected with puromycin to generate stable miR-128, anti-miR-128 or miR control lines. We included analysis of TNPO1, which we have previously validated to be a miR-128 target [33]. The relative mRNA expression of hnRNPA1, hnRNPA2B1, hnRNPK, hnRNPL and hnRNPU was measured by qRT-PCR, see Figure 1B. hnRNPA1 mRNA were significantly reduced in cells overexpressing miR-128 relative to miR controls, see Figure 1B, top left panel. In cells where endogenous miR-128 was neutralized by anti-miR-128, hnRNPA1, hnRNPA2B1, hnRNPK and hnRNPU mRNA were significantly increased, see Figure 1B. The finding that hnRNPA1 mRNA is significantly decreased or increased corresponding to overexpression or neutralization of miR-128, respectively, suggests that the effect of miR-128 on hnRNPA1 mRNA levels is specific. As hnRNPA1 plays an important role in nuclear transport by interacting with TNPO1 and is known to interact with ORF1p [43,44,45,46,47,48,49,50,51,62], we decided to focus on the regulation of hnRNPA1 and examine if hnRNPA1 is required for miR-128-induced restriction of L1 mobilization.

### 2.2. miR-128 Reduces hnRNPA1 mRNA and Protein Levels

We next examined the effects of miR-128 on hnRNPA1 by performing and validating stable miR transductions with transient miR transduction of HeLa cells. We found that both transient and stable miR transduction of miR-128 resulted in significantly reduced hnRNPA1 levels and that miR-128 neutralization enhanced hnRNPA1 mRNA levels in both experimental conditions, relative to miR controls, see Figure 2A left panel, and also Figure 1B, top left panel. Next, we determined that miR-128 overexpressing HeLa cells showed significantly reduced hnRNPA1 protein levels and anti-miR-128 significantly enhanced hnRNPA1 protein amounts, relative to miR control HeLa cells, by western blot analysis, see Figure 2B, quantifications are shown in the right panel. Different exposures of independent biological replicates are shown for miR-128 versus anti-miR-128, and confocal analysis, see Figure 2C, correlating with hnRNPA1 mRNA levels, see Figure 2A. Next, we wished to evaluate whether the observed effect of miR-128 on hnRNPA1 levels was limited to HeLa cells. We determined that miR-128 regulates hnRNPA1 mRNA levels in a panel of cell lines, including an induced pluripotent stem cell line, a cancer-initiating cell line and three different cancer cell lines (iPSCs), colon cancer initiating cells (CCIC), breast cancer cell line (MDA-MB-231), non-small cell lung cancer line (NCI-A549) and a teratoma cell line (Tera-1). miR-128 significantly reduced hnRNPA1 in all but the lung cancer cell line and anti-miR-128 showed substantially enhanced hnRNPA1 levels in all cell lines except the teratoma cell line, see Figure 2D. Finally, as expected, miR-128 was determined to also significantly regulate hnRNPA1 protein levels in three additional cancer cell lines (A549 (lung cancer), SW620 (colon cancer) and PANC1 (pancreatic cancer)), see Figure 2E, quantifications are shown below each western blot result. Different exposures of independent biological replicates are shown for miR-128 versus anti-miR-128 experiments. Together, these results demonstrate that miR-128 regulates hnRNPA1 mRNA and protein levels in multiple cell types.

### 2.3. miR-128 Binds Directly to the CDS of hnRNPA1 mRNA

We next wished to determine if hnRNPA1 mRNA is a direct target of miR-128. When performing bioinformatics analyses of potential miR-128 binding sites in hnRNPA1 mRNA, we identified one potential 7-mer seed site in the coding DNA sequence (CDS), see Figure 3A, top panel. The CDS sequence of hnRNPA1, including the seed site, was cloned into an miR-binding site luciferase reporter construct. As a positive control construct, a 23-nt with perfect complementarity was also generated, see Figure 3B, top panel. HeLa cells were co-transfected with the hnRNPA1 binding site-encoding plasmid and either mature miR-128 or miR control mimics. Luciferase activity was significantly reduced in cells co-transfected with miR-128 and the plasmid encoding the hnRNPA1 binding site relative to those co-transfected with the miR control mimic and hnRNPA1 binding site plasmid, see Figure 3B, bottom left panel. These results indicate that miR-128 binds to the seed site located in the CDS of hnRNPA1. To further characterize the specificity of miR-128 binding to the seed site in hnRNPA1, a mutated seed site was generated and cloned into the miR-binding site luciferase reporter plasmid, see Figure 3B, top panel. HeLa cells were co-transfected with either the wild-type (WT) or miR-128 seed mutant (mutant) binding site-encoding plasmids and either mature miR-128 or miR control mimics. We verified that luciferase activity was reduced in miR-128-wildtype-hnRNPA1-induced HeLa cells, compared to HeLa cells overexpressing the miR control. This finding establishes that miR-128 indeed interacts with the wildtype hnRNPA1 mRNA sequence, resulting in the inhibition of the translation of luciferase, see Figure 3B, right panel. In contrast, miR-128-mutant-hnRNPA1 over-expressing HeLa cells were characterized by de-repressed luciferase activity, to levels similar to that of the control. This result is consistent with the idea that miR-128 does not interact with the hnRNPA1 CDS mRNA sequence in which mutations have been introduced, and thus cannot inhibit luciferase activity, see Figure 3B, right panel.

Finally, in order to evaluate whether miR-128 directly interacts with hnRNPA1 mRNA in cells, Argonaute-RNA immuno-purification (miRISC-IPs) analysis were performed, see Figure 3C. Briefly, transduced HeLa cell lines stably expressing miR-128 or anti-miR-128 were lysed and Argonaute complexes relatively enriched with miR-128 (miR-128) or depleted of miR-128 (anti-miR-128) were isolated. The argonaute complexes were then analyzed for occupancy by hnRNPA1 mRNA by qRT-PCR; if hnRNPA1 mRNA is a direct target of miR-128, occupancy should be higher in complexes enriched with miR-128 relative to those depleted of miR-128 (anti-miR-128). As expected, the relative level of hnRNPA1 mRNA was significantly lower in cells stably overexpressing miR-128 compared to cells expressing anti-miR-128, see Figure 3D, top left panel “Input”. The relative fraction of Argonaute-bound hnRNPA1 mRNA was significantly increased in cells overexpressing miR-128 relative to cells expressing anti-miR-128, see Figure 3C, top right panel “IP—hnRNPA1”. When correcting for the higher amount of hnRNPA1 mRNA in anti-miR-128 input samples, the relative fractions of Argonaute-bound hnRNPA1 mRNA in IP samples were found to be more significantly reduced, relative to miR-128 samples, see Figure 3D, top right panel “corrected anti-miR-128”. miR-128 did not reduce GAPDH mRNA amounts or immuno-purified GAPDH mRNA, see Figure 3D, lower panels. These data combined, suggests that induced amounts of miR-128 result in enrichment of hnRNPA1 mRNA bound to Argonaute complexes by a direct interaction with the seed sequence in the CDS of hnRNPA1 mRNA.

### 2.4. L1 Retrotransposition Is Dependent on hnRNPA1

hnRNPA1 is a known binding partner of L1 ORF1p [51] and TNPO1 [33] suggesting a possible role for miR-128-induced regulation of hnRNPA1, affecting the L1 retro-transposition life cycle. We first wished to evaluate whether hnRNPA1 is required for successful de novo retro-transposition of L1, as previously reported [51], and secondly whether miR-128-induced L1 restriction is dependent on reduced hnRNPA1 levels.

Next, we evaluated de-novo retro-transposition activity in miR-modulated HeLa cells by performing colony formation assays. Specifically, we took advantage of previously generated variants of the neomycin reporter construct, which encodes the L1 mRNA (full-length) and a retro-transposition indicator cassette. The wild-type (WT) L1 construct is generated to encode the neomycin gene, which is inserted in the antisense direction, as it relates to a full-length L1 element. An intron has been inserted in the sense direction of the full-length L1 element. This strategy ensures that the neomycin (neo) protein can only be translated into a functional enzyme and cells can survive if L1 transcription takes place and is followed by splicing of the L1 mRNA, which is then reverse transcribed into DNA and the spliced DNA variant is then integrated back into the genome. This assay, therefore, allows for the quantification of cells with new (de novo) retrotransposition and genomic integrations events in culture. In addition, we have generated a miR-128-resistant variant of the L1 plasmid, by introducing a silent mutation into the miR-128 binding site (in the ORF2 sequence) attenuating miR-128 binding, but allowing L1 to retrotranspose (as described in [32,33]). Finally, a third variant of the L1 plasmid described by [63] encodes an L1 RNA containing a D702A mutation in the reverse transcriptase (RT) domain of the ORF2 protein, rendering the encoded L1 RT deficient (RT dead). The RT dead plasmid variant was used as a negative control in order to demonstrate that colonies obtained upon wild-type L1 plasmid transfections and neo selection are the consequence of a round of de novo L1 retro-transposition, as previously described [33]. We generated stable HeLa cell lines overexpressing full-length hnRNPA1 or plasmid control, shRNA against hnRNPA1 or a control sequence (GFP, Green Fluorescent Protein). Induced versus reduced hnRNPA1 expression levels were verified by western blot analysis, see Figure 4, right panels. hnRNPA1-modulated HeLa cell lines were then transfected with WT L1 or RT deficient L1 (RT-dead) neomycin reporter and selected for 14 days with neomycin replenished daily. We observed a significant increase in neomycin-resistant colonies in cells overexpressing hnRNPA1, relative to control HeLa cells, see Figure 4A. In contrast, cells expressing sh-hnRNPA1 were characterized by a significantly decreased number of neo-resistant colonies, compared to control HeLa cells, see Figure 4B. Importantly, hnRNPA1-modulated HeLa cells encoding the negative control (RT-dead) did not generate any neomycin-resistant colonies. These experiments demonstrate that hnRNPA1 is required for optimal L1 activity, agreeing with previous findings [51].

### 2.5. miR-128-Induced L1 Restriction Is Partly Dependent on hnRNPA1

At this point, our findings show that L1-mobilization is dependent on repressing the cellular co-factor hnRNPA1, TNPO1, and direct binding to L1 RNA. In order to evaluate the relative importance of miR-128-induced hnRNPA1 repression, we next performed rescue experiments. In brief, we generated miR-128 or miR control HeLa cell lines in which we co-expressed either full-length hnRNPA1 or plasmid controls. All HeLa cell lines were then transfected with the mutant L1 (miR-128 resistant) plasmid or the RT-dead L1 plasmid as previously described [33]. As expected, miR-128 significantly reduced L1 retrotransposition, as determined by reduced neo resistant colonies, see Figure 5. When reintroducing hnRNPA1 into miR-128 over-expressing HeLa cells, we observed a partial but significant rescue of miR-128 induced L1 restriction, see Figure 5, whereas all HeLa cell lines transfected with the RT dead L1 plasmid, resulted in no colonies following neo selection.

These results support the idea that miR-128 functions through direct binding of L1 RNA and by regulating at least two cellular co-factors (hnRNPA1 and TNPO1), which L1 is dependent on for successful mobilization.

## 3. Discussions

Interactions between hnRNPA1 and TNPO1, as well as hnRNPA1 and L1 ORF1p, have been reported [43,44,45,46,47,48,49,51,62]. Our results demonstrate that miR-128 functions to repress hnRNPA1 as one of the cellular co-factors in the L1 retro-transposition pathway and complement our earlier work describing miR-128-induced direct repression of L1 retrotransposons [32] and miR-128-induced repression of a nuclear import factor of L1 (TNPO1) [33]. This body of work agrees with the concept that miRs repress multiple cellular targets in unison to regulate important cellular pathways [34,35,36,37]. In addition, our overlay analysis indicates that miR-128 also regulates the levels of another heterogeneous nuclear ribonucleoprotein family member hnRNPL, see Figure 1, which has been reported to be required for successful L1 retro-transposition by Goodier et al. [51].

Our findings also support the conclusion that miR-128 significantly reduces hnRNPA1 levels by directly interacting with an miR-128 seed site in the coding region sequence (CDS) of hnRNPA1 mRNA. The finding that miR-128 targets the CDS of hnRNPA1 resulting in significantly reduced hnRNPA1 mRNA and, in particular, hnRNPA1 protein levels in a small panel of different cell types, was surprising. However, it is well established that miRs regulate gene products by preferentially interacting with the 3′UTR and/or the CDS of target mRNA and that such interactions are of functional importance [32,64,65]. In addition, we have determined that hnRNPA1 is required for successful miR-128-induced inhibition of L1 activity, as overexpression of hnRNPA1 in miR-128 HeLa cells can partially rescue the inhibitory effect of miR-128 on L1 mobilization. The idea that miR-128 targets multiple co-factors which L1 is dependent on at different stages in its life cycle is an area of active investigation in our laboratory and includes analysis of miR-128-induced regulation of other recently reported L1 co-factors by Mita et al. and Taylor et al. [66,67].

Mechanistic studies are also needed to dissect what exact role hnRNPA1 plays in L1 retrotransposition. It is tempting to speculate that hnRNPA1 and TNPO1 cooperate at facilitating active nuclear import of some L1-RNP complexes, and thus our finding that miR-128 target both hnRNPA1 and TNPO1 mRNAs support a possible role for miR-128 in determining whether L1-RNP complexes gain access to host DNA, independently of cell division. hnRNPA1 is also involved in many other processes necessary for cellular function including proliferation [68], mRNA splicing [69], and telomerase activity [70], and mutations in hnRNPA1 have been associated with human diseases including amyotrophic lateral sclerosis 20 [71], lung adenocarcinoma [72] and HIV-1 [73,74]. We did not observe any significant changes in cell cycle or cell toxicity when overexpressing or knocking down hnRNPA1 levels. However, due to the wide activity of hnRNPA1 in nuclear shuttling, the broader action of this protein on nuclear transport of cellular RNPs—not specific to L1, should also be taken into account. It is possible that an overall change in nuclear transport, caused by modulation of hnRNPA1 also affects L1 mobilization.

In summary, we have identified hnRNPA1 as a novel miR-128 target. We have determined that miR-128 significantly reduces hnRNPA1 protein and mRNA amounts by directly interacting with the coding sequence of hnRNPA1 mRNA and that hnRNPA1 is required for miR-128-induced L1 repression.

## 4. Materials and Methods

### 4.1. Cell Culture

All cells were cultured at 37 °C and 5% CO_2_ and routinely checked for mycoplasma (LT07-218, Lonza, Alpharetta, GA, USA). HeLa cells (CCL-2, ATCC, Manassas, VA, USA) were cultured in EMEM (SH3024401, Hyclone, Thermo Fisher Scientific, Waltham, MA, USA) supplemented with 10% HI-FBS (FB-02, Omega Scientific, Tarzana, CA, USA), 5% Glutamax (35050-061, Thermo Fisher Scientific), 3% HEPES (15630-080, Thermo Fisher Scientific), and 1% Normocin (ant-nr-1, Invivogen/Thermo Fisher Scientific). 293T cells (CRL-3216, ATCC) used to generate lentiviruses, H23 cells (CRL-5800, ATCC), and MDA-MB-231 (ATCC HTB-26) were cultured in DMEM supplemented with 10% HI-FBS (FB-02, Omega Scientific), 5% Glutamax (35050-061, Life Technologies/Thermo Fisher Scientific) and 1% Normocin (ant-nr-1, Invivogen, San Diego, CA, USA). Tera-1 cells (HTB-105, ATCC) were cultured in McCoy’s 5A (16600-082, Life Technologies, Carlsbad, CA, USA) supplemented with 20% Cosmic Serum (SH3008702, Thermo Fisher Scientific). Passaging was performed at 80% confluence with 0.25% trypsin (SH30042.01, Hyclone, Thermo Fisher Scientific). Colon cancer initiating cells (CCICs, gifted from Professor Marian Waterman, UC Irvine, CA, USA). CCICs were cultured as spheres in ultra-low attachment flasks in DMEM/F12, N2 supplement (17502-048, Life Technologies), B27 supplement (17504-044, Lifetech), heparin (4 µg/mL, Sigma-Aldrich/EMD Millipore, Burlington, MA, USA), epidermal growth factor (20 ng/mL), and basic fibroblast growth factor (20 ng/mL). H23 (CRL-5800, ATCC) were cultured in RPMI-1640 (11875, Life technologies), 10% HI-FBS, 5% Glutamax, and 1% Normocin. mESCs were maintained in knockout DMEM medium (Invitrogen) supplemented with 15% fetal bovine serum, LIF and 2i (PD0325901 and CHIR99021) as per standard techniques.

### 4.2. Transfection and Transduction

OptiMem (31985070, Life Technologies) and Lipofectamine RNAiMAX (13778075, Life Technologies) were used to complex and transfect 20 µM miR-128 mimic, anti-miR-128 or control mimics (C-301072-01 and IH-301072-02, Dharmacon, Lafayette, CO, USA) into cells. OptiMem and Xtreme Gene HP (06366236001, Roche Life Science, Penzburg, Germany) were used to transfect pJM101 neomycin L1 reporter plasmid into HeLa cells. Cells were transduced with high titer virus using polybrene (sc-134220, Santa Cruz Biotechnology, Dallas, TX, USA) and spinoculation (800 *g* at 32 °C for 30 min). Transduced cells were then selected and maintained using 3 µg/mL puromycin.

### 4.3. RNAi Using shRNA against hnRNPA1

shRNA for hnRNPA1 was designed using the RNAi Consortium (https://www.broadinstitute.org/rnai/public/) using clone TRCN0000235098 and cloned into pLKO.1 puro backbone (Addgene, #8453, Watertown, MA, USA). pLKO shGFP control plasmid was pre-assembled (Addgene, #30323).

### 4.4. Lentiviral Packaging

Lentiviral vectros (VSVG-pseudotyped) were generated by transfecting 0.67 µg of pMD2-G (12259, Addgene), 1.297 µg of pCMV-DR8.74 (8455, Addgene), and 2 µg of mZIP-miR-128, mZIP-anti-miR-128, pLKO-shControl or pLKO-shHNRNPA1 (transfer plasmid)) into 293 T cells using Lipofectamine LTX with plus reagent (15338030, ThermoFisher). Supernatants containing virus were harvested after 48 h and 96 h post-transfection. PEG-it virus precipitation solution (LV810A-1) was utilized to concentrate viral supernatants, following manufacturer’s instructions.

### 4.5. RNA Extraction and Quantification

RNA was extracted using Trizol (15596-018, ThermoFisher) and a Direct-zol RNA isolation kit (R2070, Zymo Research, Irvine, CA, USA). cDNA was made with High-Capacity cDNA Reverse Transcription Kit (4368813, ThermoFisher). Amount of hnRNPA1 mRNA was analyzed by qRT-PCR (Sense primer 5′-aagcaattttggaggtggtg-3′; Antisense primer 5′-atagccaccttggtttcgtg-3′) using Forget-me-not qPCR mastermix (Biotium, Fremont, CA, USA) relative to beta-2-microglobulin (B2M, Sense primer 5′-ATGTCTCGCTCCGTGGCCTTAGCT-3′; Antisense primer 5′-TGGTTCACACGGCAGGCATACTCAT-3′). All RT-qPCR was performed in technical cDNA and qPCR duplicates using beta-2-microglobulin (B2M) as the reference gene, based on previous analysis establishing that B2M is stably expressed in the analyzed cell lines [32,33,64]. All data were analyzed using NormFinder to ensure stability of the reference genes. For each sample, relative quantities were calculated as 2^−**Δ**Ct^ and determined as the average relative quantities in the cDNA synthesis duplicates.

### 4.6. Western Blotting

Rabbit anti-human hnRNPA1 antibody (K350, Cell Signaling Technology, Danvers, Ma, USA) was used at 1:2000. Anti-alpha Tubulin antibody (ab4074, Abcam, Cambridge, MA, USA) was diluted 1:5000 and used as a loading control, validation of antibodies can be found on the manufacturer websites. Secondary HRP-conjugated anti-rat (ab102172, Abcam) or HRP-conjugated anti-rabbit (#NA934, GE lifesciences, Pittsburg, FA, USA) was used at 1:5000. ECL substrate (32106, Thermo Fisher Scientific) was added and visualized on a BioRad ChemiDoc imager. Fluorescent quantification of protein levels was done using the LiCor Odyssey SA infrared imaging system (Invitrogen). Alternatively, quantifications were performed using Image J. Values are displayed as protein levels normalized to α-tubulin levels.

### 4.7. Argonaute-RNA Immuno-Purification

Immunopurification of all Argonaute (Ago) proteins from HeLa cell extracts was carried out using the 4F9 antibody (#sc-53521, Santa Cruz Biotechnology), as previously described in [75,76]. Briefly, 10 mm plates of 80% confluent HeLa cells were washed with buffer A (20 mM Tris-HCl pH 8.0, 140 mM KCl and 5 mM EDTA) and lysed in 200 μL of buffer 2× B (40 mM Tris-HCl pH 8.0, 280 mM KCl, 10 mM EDTA, 1% NP-40, 0.2% Deoxycholate, 2× Halt protease inhibitor cocktail (Pierce, city, if any state, country), 200 U/mL RNaseout (Life Technologies) and 1 mM DTT). Adjustment of the protein concentration across samples was obtained with buffer B (20 mM Tris-HCl pH 8.0, 140 mM KCl, 5 mM EDTA pH 8.0, 0.5% NP-40, 0.1% deoxycholate, 100 U/mL Rnaseout (Life Technologies), 1 mM DTT and 1× Halt protease inhibitor cocktail (Pierce/ThermoFisher Scientific). Centrifugation of RNA-containing lysates was carried out at 16,000 *g* for 15 min at 4 °C. Following centrifugation, supernatants were incubated with 10–20 μg of 4F9 antibody conjugated to epoxy magnetic beads (M-270 Dynalbeads, Life Technologies) for 2 h at 4 °C with gentle rotation (Nutator). The beads were isolated by magnetic separation and were washed three times for 5 min with 2 mL of buffer C (20 mM Tris-HCl pH 8.0, 140 mM KCl, 5 mM EDTA pH 8.0, 40 U/mL Rnaseout (Life Technologies), 1 mM DTT and 1× Halt protease inhibitor cocktail (Pierce)). Immunopurification was performed, after which RNA was isolated using miRNeasy kits (QIAGEN, Germantown, MD, USA), according to the manufacturer’s recommendations. Finally, qRT-PCR was carried out using hnRNPA1 primers designed around the binding site of miR-128 (Sense primer 5′-TCTCCTAAAGAGCCCGAACA-3′; Antisense primer 5′-TTGCATTCATAGCTGCATCC-3′) or GAPDH (Sense primer 5′-GGTGGTCTCCTCTGACTTCAA-3′; Antisense primer 5′-GTTGCTGTAGCCAAATTCGTT-3′) normalized to B2m (Sense primer 5′-ATGTCTCGCTCCGTGGCCTTAGCT-3′; Antisense primer 5′-TGGTTCACACGGCAGGCATACTCAT-3′). Results were normalized to their inputs.

### 4.8. Cloning

To generate the hnRNPA1, full-length clone modifications of the pFC-PGK-MCS-pA-EF1-GFP-T2A-Puro plasmid (SBI, backbone, Palo Alto, CA, USA) were carried out by specifically replacing the PGK with a CMV (cytomegalovirus) promoter. The CMV promoter, which is a strong and robust promoter, was PCR amplified using the phiC31 integrase expression plasmid (SBI) as a template. The CMV promoter insert was generated by using sense CMV primer (5’-CTAGAACTAG TTATTAATAG TAATCAATTA CGGGGTC-3´) and antisense CMV primer (5´-GATATCGGAT CCACCGGTAC CAAGCTTAAG TTTAAAC-3´). The plasmid, containing the insert and backbone, was then cut by *Xba*I and *BamH*I and purified by agarose gel electrophoresis. The insert and backbone-containing plasmid was ligated together utilizing the Quick ligation kit (NEB) and then transformed, using Cold Fusion competent cells (1 × 10^9^ cfu/µg). Finally, the pFC-CMV-MCS-pA-EF-1-GFP-T2A-Puro-MH1 plasmid was verified by sequencing.

In order to generate full-length hnRNPA1 mRNA expression plasmid, total RNA from HeLa cells was isolated, 20 ng was reverse transcribed using poly dT primer. Phusion High-Fidelity PCR Kit (NEB, Ipswich, MA, USA) was used to generate all amplicons by following the manufacturer´s protocol. The fragments were assembled stepwise and cloned into the pFC-CMV-MCS-pA-EF-1-GFP-T2A-Puro-MH1 *Bam*HI/*Cla*I linearized backbone, using the Cold Fusion kit (SBI). The pFC-CMV-TNPO1-pA-EF-1-GFP-T2A-Puro-MH1 plasmid was verified by sequencing. FL-Control is an empty vector.

### 4.9. Luciferase Binding Assay

Wild-type hnRNPA1 sense primer (5′-AATTCTTGGGTTTGTCACATATGCCACTGTGGAGGAGGTGGATGCAGCTA-3′) and antisense primer (5′-CTAGTAGCTGCATCCACCTCCTCCACAGTGGCATATGTGACAAACCCAA-3′), mutated hnRNPA1 sense primer (5′-AATTCTTGGGTTTGTCACATATGCCCTTATGGAGGAGGTGGATGCAGCTA-3′) and antisense primer (5′-CTAGTAGCTGCATCCACCTCCTCCATAAGGGCATATGTGACAAACCCAA-3′), or positive control sense primer (5′-AATTCAAAGAGACCGGTTCACTGTGAA-3′) and antisense primer (5′-CTAGTTCACAGTGAACCGGTCTCTTTG-3′) sequences were cloned into dual-luciferase reporter plasmid (pEZX-MT05, GeneCopoeia, Rockville, MD, USA). A total of 3 × 10^5^ HeLa cells were forward transfected with 0.8 µg of reporter plasmid (WT, mutated, Pos) and 20 nM miR-128 mimic (Dharmacon) or Control mimic (Dharmacon) using Attractene transfection reagent (301005, Qiagen), according to manufacturer instructions. Relative Gaussia Luciferase and SEAP was determined using a Secrete-Pair Dual Luminescence Assay Kit (SPDA-D010, Genecopoeia). Luminescence was detected using a Tecan Infinite F200 Pro microplate reader.

### 4.10. Site Directed Mutagenesis

A reverse transcriptase incompetent (PJM101/L1) plasmid control was generated by following the published mutagenesis strategy of Morrish et al., utilizing the Q5 Site-directed mutagenesis Kit (E0554S, NEB). Specifically, the D702A mutation in L1 ORF2 was carried out, resulted in a non-functional reverse transcriptase enzyme, referred to as, RT-dead L1 [77].

### 4.11. Colony Formation Assay

Stable HeLa lines expressing miR-128, anti-miR-128, scramble control, shControl, shhnRNPA1, shTNPO1, Full-length hnRNPA1 or TNPO1, or plasmid control were transfected with pJM101/L1RP or RT-dead L1 plasmid (containing neomycin resistance retrotransposition indicator cassette) per well using X-treme gene HP DNA transfection reagent (06366236001, Roche, Basel, Switzerland) according to manufacturer instructions. Cells were selected using 500 µg/mL G418 (ant-gn-1, Invivogen). Neomycin-resistant colonies were fixed with cold 1:1 methanol:acetone, then visualized using May-Grunwald (ES-3410, Thermo Fisher Scientific) and Jenner-Giemsa staining kits (ES-8150, Thermo Fisher Scientific) according to manufacturer’s protocol. Selection began with 25 µg/mL G418 72 h post-transfection and selection was maintained with daily media changes until negative control (non-transfected) cells died. Neomycin-resistant colonies were fixed as described above.

### 4.12. RNA Sequencing and Data Analysis

DGCR8^−/−^ mESCs were transfected with miR controls or miR-128 in triplicate cultures and harvested cells after 12 h in order to enrich for primary target mRNAs, as opposed to studying secondary effects of miR-128. Two replicates of each triplicate were selected and cDNA libraries were generated using the Smart-seq2 protocol [58]. The libraries were sequenced as 43 bp paired-end reads. STAR [59] was used to align the reads on to the mm9 genome. RSEM [60] was used to quantitate the gene expression and EBSeq [61] was used to identify differentially expressed genes.

## 5. Conclusions

Our results demonstrate that miR-128 represses L1 mobilization through a multi-facetted mechanism by both directly targeting of L1 RNA and indirectly through the repression of cellular co-factors, possibly a network of co-factors which L1 is dependent on, including TNPO1 and hnRNPA1 [32,33].

## Figures and Tables

**Figure 1 ijms-20-01955-f001:**
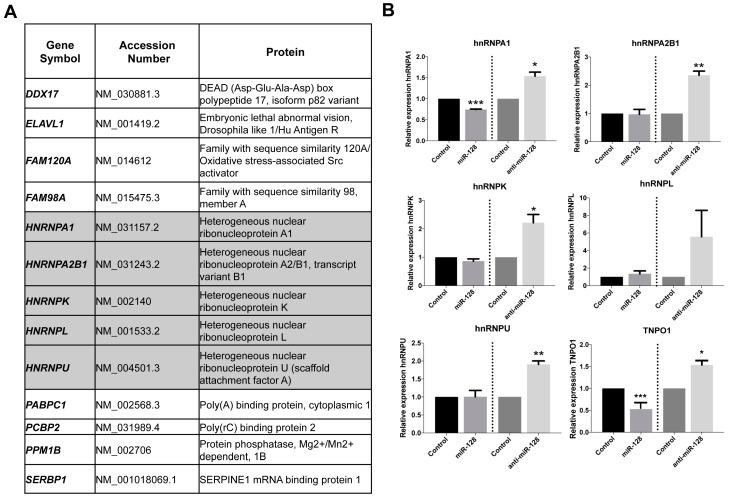
Identification of hnRNPA1 as a cellular target of miR-128. (**A**) Table showing the results of an overlay analysis of the genes identified in our mouse embryonic stem cell (mESC) DGCR8^−/−^ screen of putative miR-128 targets by differential gene expression and protein targets previously reported as L1 ORF1p interactome [5]. Members of the heterogeneous nuclear ribonucleoproteins (hnRNPs) family of RNA-binding proteins are highlighted in grey, see Appendix A for additional information. (**B**) Relative amount of hnRNPA1, hnRNPA2B1, hnRNPK, hnRNPL and hnRNPU and TNPO1 (positive control) mRNA normalized to B2M determined in HeLa cells stably transduced with control miR, anti-miR-128 or miR-128 are shown as mean ± SD (*n* = 3 technical replicates, * *p* < 0.05, ** *p* < 0.01, *** *p* < 0.001).

**Figure 2 ijms-20-01955-f002:**
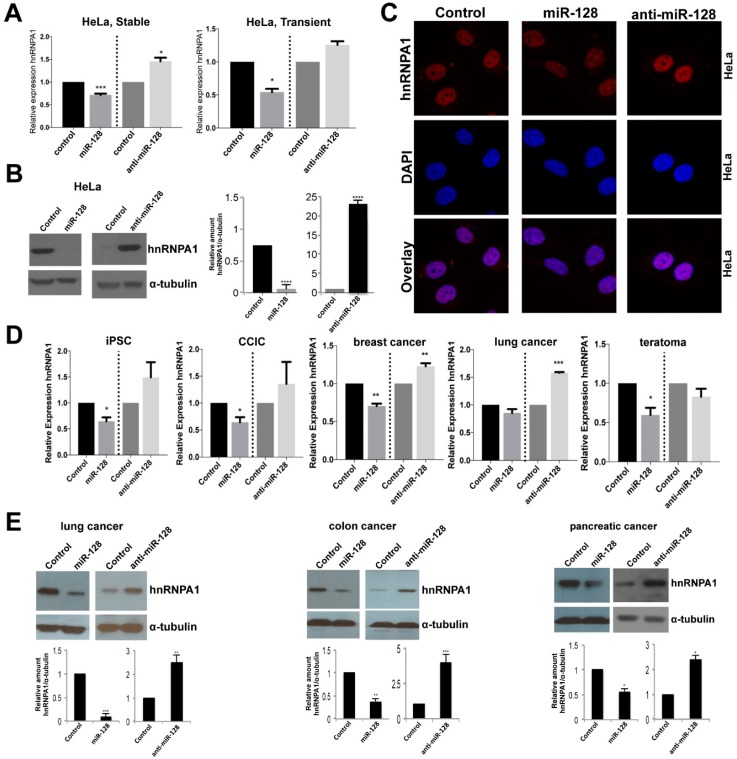
miR-128 reduces hnRNPA1 mRNA and protein amounts whereas miR-128 neutralization enhances hnRNPA1 expression levels in multiple cell types. (**A**) Relative amount of hnRNPA1 mRNA normalized to B2M in HeLa cells stably transduced with miR-128, anti-miR-128 or control constructs (left panel, *n* = 2 independent biological replicates, *p* = ns); or transiently transfected with miR-128, anti-miR-128 or control mimics (right panel, mean ± SEM, *n* = 3 independent biological replicates, * *p* < 0.05) (**B**) Immunoblot analysis of hnRNPA1 and α-tubulin protein levels in lysates from HeLa cells transduced with miR-128, anti-miR-128 or miR control constructs (left panel). Quantification of blots is shown (right panel). (**C**) Stable miR-128, anti-miR-128 and control miR HeLa cell lines were analyzed by immunofluorescence for hnRNPA1 expression and co-localization with DAPI. (**D**) Relative amounts of hnRNPA1 mRNA normalized to B2M in induced pluripotent stem cells, colorectal cancer initiating cells (CCIC), breast cancer cells (MDA-MB-231), non-small cell lung cancer (A549) cells and teratoma (Tera-1) cells. (**E**) Immunoblot analysis of hnRNPA1 and α-tubulin protein levels in protein-containing lysates isolated from non-small cell lung cancer (A549), colon cancer (SW620) (PANC1) cells transduced with miR-128, anti-miR-128 or miR control constructs. Quantification of blots are shown (bottom panels). Throughout the figure if not otherwise noted, *n* = 3 independent biological replicates, mean ± SEM, * *p* < 0.05, ** *p* < 0.01, *** *p* < 0.001, **** *p* < 0.0001, calculated by students t-test.

**Figure 3 ijms-20-01955-f003:**
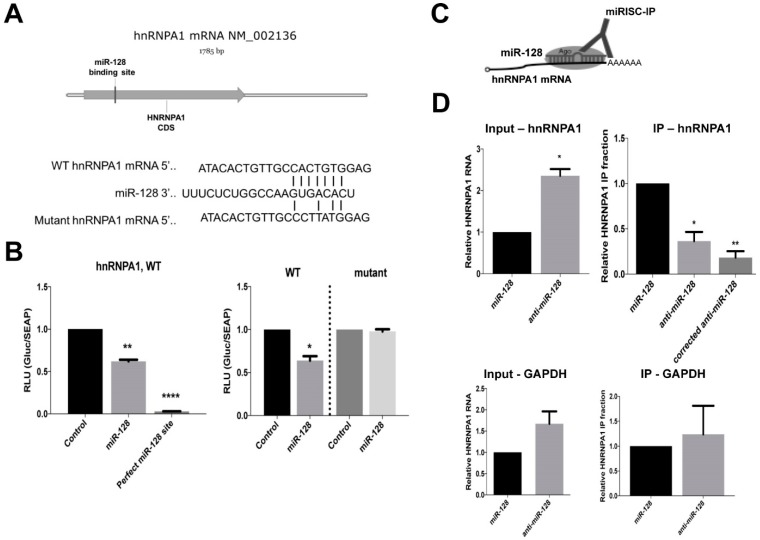
miR-128 represses hnRNPA1 expression by binding directly to coding sequence (CDS) RNA. (**A**) Schematic of the predicted miR-128 7-mer binding site in the coding region (CDS) of hnRNPA1 mRNA (top panel). (**B**) Cartoon showing the predicted base pairing of miR-128 to the seed sequence of wild-type (WT) hnRNPA1 as well as a representation of mutations in the seed sequence (mutant) used for luciferase binding assays (top panel). Relative luciferase activity in HeLa cells transfected with plasmids expressing a Gaussia luciferase gene fused to the wild-type (WT) binding site or positive control sequence corresponding to the 22 nucleotide perfect match of miR-128 and co-transfected with control or mature miR-128 mimics were determined 48 h post-transfection (bottom left panel, n = 3 independent biological replicates, mean ± SEM, ** *p* < 0.01, **** *p* < 0.0001). Relative luciferase activity in HeLa cells transfected with plasmids expressing the luciferase gene fused to the WT or the mutated binding site (mutant) and co-transfected with control or mature miR-128 mimics were determined 48 h post-transfection (bottom right panel, *n* = 3 independent biological replicates, mean ± SEM, * *p* < 0.05). (**C**) Cartoon of Argonaute-RNA immuno-purification (miRISC-IP). (**D**) miRISC IP of HeLa cell lines stably transduced with miR-128 overexpression or miR-128 neutralization (anti-miR-128) were performed. Relative amounts of hnRNPA1 RNA normalized to B2M were determined for input samples (top left panel “input—hnRNPA1”, *n* = 3 independent biological replicates, mean ± SEM, * *p* < 0.05). Relative fractions of hnRNPA1 transcript amounts associated with immune-purified Ago complexes are shown for immunopurified (IP) samples, hnRNPA1 fractions normalized to the amount of TNPO1 in the input are shown as “corrected” (top right panel “IP—hnRNPA1”, *n* = 3 independent biological replicates, mean ± SEM, * *p* < 0.05, ** *p* < 0.01). (**D**) Relative amount of GAPDH in the same input and IP samples were determined as a negative control (top right panel “IP—hnRNPA1”, *n* = 3 independent biological replicates, mean ± SEM). Throughout the figure, *n* = 3 independent biological replicates, mean ± SEM, * *p* < 0.05, ** *p* < 0.01, **** *p* < 0.0001, calculated by students t-test.

**Figure 4 ijms-20-01955-f004:**
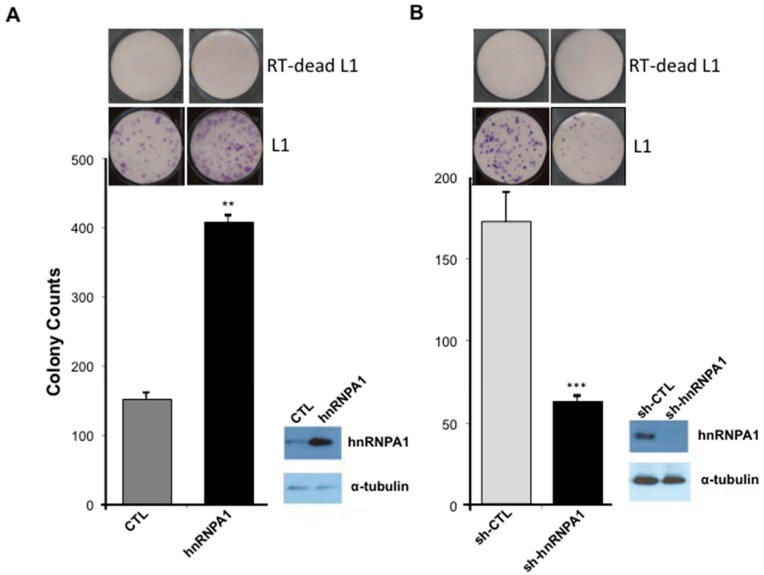
Overexpression of hnRNPA1 enhances de novo L1 retro-transposition and hnRNPA1 knock-down reduces L1 mobilization. (**A**) De novo retro-transposition was determined by a colony formation assay in HeLa cells stably transduced with plasmids encoding control plasmid (control), hnRNPA1 and transfected with L1 expression construct. Western blot analysis validating reduced levels of hnRNPA1 in over-expressing cell lines are shown. α-tubulin was used as a loading control (right panel). (*n* = 3 technical replicates, mean ± SD, ** *p* < 0.01). (**B**) Representative example of neomycin-resistant colony counts from a colony formation assay in HeLa cells stably transduced with plasmids encoding shRNA against GFP (Green Fluorescent Protein) (control), hnRNPA1 and transfected with WT L1 (L1) or RT deficient L1 (RT-dead L1) expression construct. (*n* = 3 technical replicates, mean ± SD, *** *p* < 0.001). Western blot analyses of hnRNPA1 and α-tubulin in knock-down HeLa cell lines are shown (right panels).

**Figure 5 ijms-20-01955-f005:**
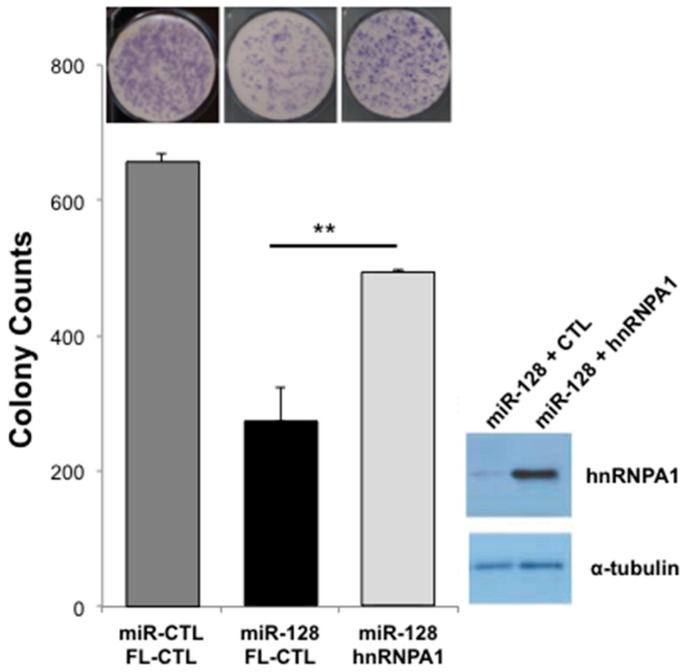
hnRNPA1 partly rescues miR-128-induced inhibition of de novo L1 retrotransposition. The functional importance of hnRNPA1 in miR-128-induced L1 repression was evaluated by colony formation assays using mutant L1 (miR-128 resistant) or RT deficient L1 (RT-dead L1) expression constructs in stable HeLa cell lines expressing either miR-control (miR-CTL) or miR-128 along with either a plasmid control (FL-CTL) or induced hnRNPA1 (hnRNPA1) (*n* = 3 technical replicates, mean ± SD, ** *p* < 0.01). Western blot analysis was performed for hnRNPA1 and α-tubulin to validate increased levels of hnRNPA1 in miR-128-hnRNPA1 rescue HeLa cell lines (panel).

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
