# Peer review of "miR-128 Restriction of LINE-1 (L1) Retrotransposition Is Dependent on Targeting hnRNPA1 mRNA"

_ijms, 2019, doi:10.3390/ijms20081955_

Round 1
Reviewer 1 Report
In this manuscript, Fung et al. identified hnRNPA1, which is a cellular co-factor for LINE-1 (L1) retrotransposition, to be an additional target for miR-128-mediated repression. This work is a continuation of previous work reported in 2015 and 2017 (ref. 32 and 33), where authors showed that miR-128 represses L1 retrotransposition by direct binding to L1 RNA and also by down-regulating the nuclear import factor TNPO1. Altogether these results describe how miR-128 involved in regulation of L1 retrotransposition.
I have several concerns regarding experiments. First, the description of qRT-PCR method is insufficient. For example, there is no explanation for why beta-2-microglobulin was chosen as a reference gene. I believe that the validation of reference genes and qPCR conditions (also absent) was performed before but there is no reference to where it can be found. I recommend adjusting this section according to MIQE guidelines (Bustin et al., 2009). Since the level of hnRNPA1 mRNA in HeLa cells overexpressing miR-128 was reduced only slightly (Figure 1B), it is important to provide the complete method details due to technical limitations of ΔΔCt qPCR. Also, it would be useful if authors would clarify how exactly quantification of immunoblots have been done (e.g. data shown in Figure 2B).
Overall, the manuscript is well written. Some minor language and style editing are still required. For example, the phrase from Abstract (p.1 line 28) “Finally, we determine that hnRNPA1 is a functional target of miR-128 and that induced hnRNPA1 expression in miR-128-overexpressing cells can partly rescue the miR-128-induced repression of L1’s ability to transpose to different genomic locations.” is too long and difficult to understand unless you read it for the second time. The phrase on p.4 (lines 22-25) “We determined that miR-128 regulates hnRNPA1 mRNA levels in an induced pluripotent stem cell line, in a cancer initiating cell line and in three different cancer cell lines, and (iPSCs, colon cancer initiating cells (CCIC), breast cancer cell line (MDA-MB-231), non-small cell lung cancer line (NCI-A549) and a teratoma cell line (Tera-1).” also requires correction. The font and/or quality of Figure 2 should be increased as it’s hard to read labels. The section “Materials and methods” requires moderate editing (see below).
Additional comments:
p.1 line 12. In my opinion, the word “Surprisingly” is unnecessary here.
p.2 line 37. “38aa” should be “38 aa”.
p.3 line 27. “Fig1B” should be “Fig 1B”. Same for p.4 line 16.
p.5 line 25. “23nt” should be “23-nt”.
p.7 line 49. “Fig4A” should be “Fig 4A”.
In several places, the space is missing between numbers and units (e.g. p.10 lines 35, 41, 45; p.11 lines 5, 6, 27, 29; p.12 lines 26, 41; p. line 2). Also, in several places again authors used “hr” instead of “hours” or “h” as it should be.
Please, use the symbol “µ” or “u” uniformly.
p.10 line 44. “800xg” should be “800 g”
p.11 lines 34 and 36. The degree sign should be fixed.
p.11 line 37. “five mins” should be “5 minutes” or “5 min”.
p.11 line 40. “hn RNPA1” should be “hnRNPA1”.
p.12 line 8. “quick ligation kit” should be “Quick Ligation™ Kit”. Same for line 13 “cold fusion kit”, as it refers to the product’s name. Also, please specify what cells were used for transformation.
Supplementary data file: in the list “Control vs. miR128 Expt 1” lines 1069 and 1070 display data (15.09.2015 and 02.09.2015, respectively) instead of genes names. Same for the list “Control vs. miR128 Expt 2” line 479 (05.03.2015). What does it mean?
Author Response
Thank you for the helpful comments. Please find my answers below in “blue and Italic”.
I have several concerns regarding experiments. First, the description of qRT-PCR method is insufficient. For example, there is no explanation for why beta-2-microglobulin was chosen as a reference gene. I believe that the validation of reference genes and qPCR conditions (also absent) was performed before but there is no reference to where it can be found. I recommend adjusting this section according to MIQE guidelines (Bustin et al., 2009). Since the level of hnRNPA1 mRNA in HeLa cells overexpressing miR-128 was reduced only slightly (Figure 1B), it is important to provide the complete method details due to technical limitations of ΔΔCt qPCR. Also, it would be useful if authors would clarify how exactly quantification of immunoblots have been done (e.g. data shown in Figure 2B). This information has now been included in the Method section on page 16, and the added sections are included below.
“All RT-qPCR was performed in technical cDNA and qPCR duplicates using beta-2-microglobulin (B2M) as reference gene, based on previous analysis establishing that B2M is stably expressed in the analyzed cell lines [32, 33, 67]. All data was analyzed using NormFinder to ensure stability of the reference genes. For each sample, relative quantities were calculated as 2−ΔCt and determined as the average relative quantities in the cDNA synthesis duplicates”.
“Fluorescent quantification of protein levels was done using the LiCor Odyssey SA infrared imaging system (Invitrogen). Alternatively quantifications were performed using Image J. Values are displayed as protein levels normalized to α-tubulin levels.
Overall, the manuscript is well written. Some minor language and style editing are still required. For example, the phrase from Abstract (p.1 line 28) “Finally, we determine that hnRNPA1 is a functional target of miR-128 and that induced hnRNPA1 expression in miR-128-overexpressing cells can partly rescue the miR-128-induced repression of L1’s ability to transpose to different genomic locations.” is too long and difficult to understand unless you read it for the second time. The phrase on p.4 (lines 22-25) “We determined that miR-128 regulates hnRNPA1 mRNA levels in an induced pluripotent stem cell line, in a cancer initiating cell line and in three different cancer cell lines, and (iPSCs, colon cancer initiating cells (CCIC), breast cancer cell line (MDA-MB-231), non-small cell lung cancer line (NCI-A549) and a teratoma cell line (Tera-1).” also requires correction. The sentences mentioned have been rewritten in the Abstract page 1 and in the Results section on page 7 and is included below.
“Furthermore, we establish that hnRNPA1 is a functional target of miR-128. Finally, we determine that induced hnRNPA1 expression in miR-128-overexpressing cells can partly rescue the miR-128-induced repression of L1’s ability to transpose to different genomic locations”.
“We determined that miR-128 regulates hnRNPA1 mRNA levels in a panel of cell lines, including an induced pluripotent stem cell line, a cancer initiating cell line and three different cancer cell lines (iPSCs, colon cancer initiating cells (CCIC), breast cancer cell line (MDA-MB-231), non-small cell lung cancer line (NCI-A549) and a teratoma cell line (Tera-1)”.
The font and/or quality of Figure 2 should be increased as it’s hard to read labels.
Figure 2 has been updated and is attached.
The section “Materials and methods” requires moderate editing (see below).
Additional comments:
All the mentioned edits have been corrected/included.
p.1 line 12. In my opinion, the word “Surprisingly” is unnecessary here.
p.2 line 37. “38aa” should be “38 aa”.
p.3 line 27. “Fig1B” should be “Fig 1B”. Same for p.4 line 16.
p.5 line 25. “23nt” should be “23-nt”.
p.7 line 49. “Fig4A” should be “Fig 4A”.
In several places, the space is missing between numbers and units (e.g. p.10 lines 35, 41, 45; p.11 lines 5, 6, 27, 29; p.12 lines 26, 41; p. line 2). Also, in several places again authors used “hr” instead of “hours” or “h” as it should be.
Please, use the symbol “µ” or “u” uniformly.
p.10 line 44. “800xg” should be “800 g”
p.11 lines 34 and 36. The degree sign should be fixed.
p.11 line 37. “five mins” should be “5 minutes” or “5 min”.
p.11 line 40. “hn RNPA1” should be “hnRNPA1”.
p.12 line 8. “quick ligation kit” should be “Quick Ligation™ Kit”. Same for line 13 “cold fusion kit”, as it refers to the product’s name. Also, please specify what cells were used for transformation.
We used the included Cold Fusion competent cells (1x109 cfu/µg), which comes with the kit. This information is included in the Method section on page 18.
Supplementary data file: in the list “Control vs. miR128 Expt 1” lines 1069 and 1070 display data (15.09.2015 and 02.09.2015, respectively) instead of genes names. Same for the list “Control vs. miR128 Expt 2” line 479 (05.03.2015). What does it mean?
This has been corrected. The gene names in specific rows (Expt 1 rows 684, 1069, 1070 and Expt 2 row 479) were displayed as dates due to Microsoft Excel's automatic conversion of the gene names. For example, the gene name SEPT15 was converted to a date. We have formatted those cells in excel to display the data as text to prevent this conversion. The updated Supplemental Figure 1 is attached.
Reviewer 2 Report
L1s are elements of human genome taking part in generation of genomic diversity. They can also provide genomic instability and various diseases. The manuscript “miR-128-induced LINE-1 restriction is dependent on down-regulation of hnRNPA1” is devoted to study of mechanisms restrict L1 transpositions. The authors showed miR-128-induced restriction of L1 transpositions. Furthermore, the interactions between hnRNPA1 and TNPO1, as well as hnRNPA1 and L1 ORF1p. They showed that miR-128 represses hnRNPA1 as one of the cellular co-factors in the L1 retrotransposition pathway. Some of these results are not quite new. Some facts were described earlier. However, these results are some additional data required for confirmation of earlier obtained ones. In general, I suppose that results obtained in this study are important for our understanding of suppression of L1 retrotranspositions.
Small remarks to the text of manuscript:
The authors refer L1 to genomic parasites. From my point of view, it would be better to avoid the term ‘parasites’.
Many times, they wrote about ‘L1 life cycles’. It is not quite clear what that means.
The abbreviation L1-RNP for L1 Ribonuclear Protein was introduced in Line 3 (Page 3) but it was use earlier ( Line 40 and 49, Page 2).
There was no information on origin of DGCR8-/- mESCs.
Author Response
Thank you for the helpful comments. Please find my answers below in “blue and Italic”.
The authors refer L1 to genomic parasites. From my point of view, it would be better to avoid the term ‘parasites’. We have corrected the term “parasite” in the Abstract on page 1 to “transposable elements”.
Many times, they wrote about ‘L1 life cycles’. It is not quite clear what that means. Good point, now detailed in the Introduction section, on page 2. The addition is also included below.
“The L1 life cycle is dependent on replicating the L1 element using a “copy and paste” mechanism with an RNA intermediate [11, 12]”.
The abbreviation L1-RNP for L1 Ribonuclear Protein was introduced in Line 3 (Page 3) but it was use earlier (Line 40 and 49, Page 2). This has been fixed.
There was no information on origin of DGCR8-/- mESCs. This information has now been included in the Result section on page 5 and is included below.
The Dgcr8 knockout ES cells were generated by removing exon 3, resulting in the formation of several premature stop codons downstream of the targeted region.